# Optimizing Precision: A Trajectory Tract Reference Approach to Minimize Complications in CT-Guided Transthoracic Core Biopsy

**DOI:** 10.3390/diagnostics14080796

**Published:** 2024-04-10

**Authors:** Stella Chin-Shaw Tsai, Tzu-Chin Wu, Frank Cheau-Feng Lin

**Affiliations:** 1Superintendent Office, Taichung MetroHarbor Hospital, Taichung 43503, Taiwan; tsaistella111@gmail.com; 2Department of Post-Baccalaureate Medicine, College of Medicine, National Chung Hsing University, Taichung 402202, Taiwan; 3Department of Pulmonary Medicine, Chung Shan University Hospital, Taichung 40201, Taiwan; ts@csh.org.tw; 4School of Medicine, Chung Shan Medical University, Taichung 40201, Taiwan; 5Department of Thoracic Surgery, Chung Shan Medical University Hospital, Taichung 40201, Taiwan

**Keywords:** CT-guide biopsy, core needle, lung, success rate, complications, pneumothorax, hemoptysis

## Abstract

The advent of computed tomography (CT)-guided transthoracic needle biopsy has significantly advanced the diagnosis of lung lesions, offering a minimally invasive approach to obtaining tissue samples. However, the technique is not without risks, including pneumothorax and hemorrhage, and it demands high precision to ensure diagnostic accuracy while minimizing complications. This study introduces the Laser Angle Guide Assembly (LAGA), a novel device designed to enhance the accuracy and safety of CT-guided lung biopsies. We retrospectively analyzed 322 CT-guided lung biopsy cases performed with LAGA at a single center over seven years, aiming to evaluate its effectiveness in improving diagnostic yield and reducing procedural risks. The study achieved a diagnostic success rate of 94.3%, with a significant reduction in the need for multiple needle passes, demonstrating a majority of biopsies successfully completed with a single pass. The incidence of pneumothorax stood at 11.1%, which is markedly lower than the reported averages, and only 0.3% of cases necessitated chest tube placement, underscoring the safety benefits of the LAGA system. These findings underscore the potential of LAGA to revolutionize CT-guided lung biopsies by enhancing procedural precision and safety, making it a valuable addition to the diagnostic arsenal against pulmonary lesions.

## 1. Introduction

Accurate diagnosis and classification of lung tumors are imperative for determining optimal treatment strategies. While resection remains a common practice, the era of minimally invasive procedures has ushered in cost-effective alternatives, particularly in regions with endemic pulmonary tuberculosis, such as Taiwan [1]. Despite these advancements, challenges persist, especially in the precise diagnosis of malignancy, notably in ground glass opacity (GGO) lesions, where traditional methods like FDG-PET exhibit limitations [2].

In the realm of evolving diagnostic methodologies, achieving precise preoperative localization becomes paramount, particularly for central lesions necessitating anatomical resection. Current biopsy techniques, ranging from bronchoscopic biopsy and endobronchial ultrasound-fine needle aspiration (FNA) to CT-guided procedures, demonstrate varying efficacy based on tumor location and institutional resources [3,4]. Bronchoscopic procedures are well-established for their high diagnostic yield in evaluating central lung lesions and represent the preferred non-surgical procedure for these cases. However, for peripheral lesions, where bronchoscopy might be less effective due to accessibility issues, CT-guided approaches have gained prominence. CT guidance facilitates precise needle placement, ensuring avoidance of damage to surrounding structures and extending the diagnostic capabilities to areas beyond the reach of bronchoscopy [5]. In the genomic era, there is a growing need for substantial tissue samples to meet the demands of genetic studies [6]. Core biopsy, with its superior capabilities compared to FNA, necessitates using larger biopsy needles. However, CT-guided transthoracic core biopsy (CTTCB) carries notable complication rates [7,8], including pneumothorax and pulmonary hemorrhage, causing hesitation among practitioners, exacerbated by low reimbursement rates [9].

In response to these challenges, the Laser Angle Guide Assembly (LAGA) system emerges as an innovative solution designed to augment the precision and safety of CTTCB. The LAGA system comprises a portable laser unit that projects a beam to delineate the optimal needle trajectory based on pre-procedural CT images. This guidance allows for real-time adjustment of the needle’s path, minimizing the risk of damaging surrounding structures and improving the likelihood of successful tissue sampling on the first attempt. The incorporation of the LAGA system into CT-guided biopsy procedures reflects a step forward in the pursuit of minimally invasive diagnostic techniques that prioritize patient safety and procedural efficiency. Initially crafted for biopsy procedures, LAGA has demonstrated effectiveness in pre-operative localization and pulmonary tumor ablation, leading to reduced puncture frequency and complications, notably pneumothorax [10,11].

This study endeavors to scrutinize the impact of the LAGA system on the success and complication rates of CTTCB. By juxtaposing our outcomes with conventional CT-guided biopsy literature, we aim to shed light on the potential advantages of this novel approach. We focus on addressing the dual challenges of acquiring sufficient tissue for diagnosis and genetic studies while mitigating procedure-related complications.

## 2. Materials and Methods

### 2.1. Patient Inclusion

This retrospective observational study collected data from consecutive patients who underwent CTTCB at a medical center in Taiwan between 2015 and 2021. Patient information, documented in hard copy records, was cross-referenced and confirmed using electronic medical records and Picture Archiving and Communications System (PACS).

### 2.2. CTTCB-LAGA Procedure

The intricacies of the procedures detailed in our prior publication [12] and the accompanying video demonstrating the pre-operative CT-guided hookwire localization [11] are worth noting. Briefly, the angle of the LAGA was determined based on the CT image, aligning its tip laser with the needle puncture point as seen in the video. The portable green laser level was projected onto the front index line of the LAGA, matching the anticipated angle on CT. The needle was accurately inserted at the point where the two laser lines intersected with the lesion. In the current investigation, the established protocol remained consistent across all cases. Specifically, the biopsy procedures were performed using a 16-gauge × 15 cm Temno^TM^ coaxial needle set (BD, Franklin Lakes, NJ, USA). The choice of a 16-gauge needle, larger than the commonly used 18-gauge needles for lung biopsies, was made to facilitate the acquisition of adequate tissue samples for both pathological diagnosis and genetic analysis, reflecting the evolving needs of genomic studies in the field of thoracic oncology. In addition, it is important to mention that we did not use the adjunctive approach of sealing the puncture track with gel–foam slurry in our investigation, despite its known potential benefits in decreasing complications such as pneumothorax and bleeding. The decision was made due to the absence of gel–foam slurry for pulmonary treatments in Taiwan throughout the timeframe of our investigation.

In addressing the challenge of respiratory motion during the biopsy, our strategy diverged from the conventional practice of instructing patients to hold their breath. Instead of relying on breath-holding, which could lead to variability and deviation owing to different capacities among patients, we asked them to breathe normally. We meticulously planned and executed needle insertion to coincide with the end-inspiratory or end-expiratory phase of the patient’s normal breathing cycle, depending on which phase offered the greatest stability and visibility of the target lesion. This approach was facilitated by real-time CT imaging to determine the optimal moment for needle advancement, thus ensuring precise alignment with the patient’s respiratory motion.

In this study, all CTTCB-LAGA procedures were conducted using the Philips CT Brilliance 64 system (Hi Tech International Group, Inc., Deerfield Beach, FL, USA), equipped with a specialized lung biopsy protocol designed to optimize imaging quality while minimizing patient radiation exposure. The Philips CT Brilliance 64, a state-of-the-art multi-detector CT (MDCT) scanner, features 64 detector rows capable of providing high-resolution images essential for the precise localization and targeting of lung lesions. This system supports real-time CT imaging, facilitating accurate needle placement and adjustments during the biopsy procedure.

The lung biopsy protocol implemented on the Philips CT Brilliance 64 was carefully developed to ensure optimal visualization of the target lesion and the surrounding anatomical landmarks. Key parameters of the protocol included a slice thickness of 1 mm for detailed cross-sectional images, with tube voltage and current adjusted based on the patient’s size and specific diagnostic requirements. This meticulous approach to CT acquisition allowed for the precise alignment of the biopsy needle with the target lesion, leveraging the advanced imaging capabilities of the Philips CT Brilliance 64 to enhance the safety and efficacy of the LAGA system during procedures.

Of particular consideration in our endeavors was the trajectory angle from the puncture point to the target. Addressing this concern, we devised a method involving the projection of a laser line from the caudal to the cephalic position of the patient. This laser trajectory, meticulously aligned with the calculated angle on our workstation, served as a tangible reference for the planned trajectory during the introduction of the biopsy needle (Figure 1).

### 2.3. Data Collection

We systematically documented key parameters, including gender, age, pulmonary function, tumor size, location, patient position, needle trajectory angle, depth, puncture frequency, biopsy pieces, and pathology. The success of the biopsy was defined as (1) confirmation of malignant pathology, (2) identification of meaningful benign pathology, (3) detection of other benign pathology without subsequent malignancy during follow-up, and (4) instances of biopsy failure not documented in the medical chart. Comprehensive recording extended to the evaluation of complications, encompassing pneumothorax, parenchymal hemorrhage, hemoptysis, hemothorax, pain, and cough. Pneumothorax was precisely defined as the presence of any air in the pleural space, discerned through meticulous CT examination [8]. Additionally, parenchymal hemorrhage was identified through the observation of any new ground glass opacity in the post-biopsy CT scans.

### 2.4. Statistics

Chi-square and Fisher’s exact tests were employed for categorical variables when the number in a cell was less than five. Linear-by-linear tests were utilized for ordinal variables. The normality of data was assessed using the Kolmogorov–Smirnov test. For numerical data, Student’s *t*-test was applied for variables with a normal distribution, and the Mann–Whitney U test was used for those with non-normal distribution. Multivariate analyses, specifically stepwise backward binary logistic regression, were conducted to identify factors associated with complications and success.

## 3. Results

### 3.1. Demography

A total of 322 CTTCB-LAGA procedures were included. The patient cohort was predominantly male (58.7%), with a mean age of 62 years (SD 13.6) (Table 1). The right upper lobe was the most common biopsy site (28.5%). Patient positioning during biopsies was primarily in the prone (50.3%) and supine (40.7%) positions. The median number of tissue cores obtained was 3 (range 1–7). Biopsy results revealed primary lung cancer in 66.1% of cases, metastatic disease in 5.1%, benign lesions in 14.0%, and atypical/non-diagnostic findings in 14.6%.

### 3.2. CTTCB-LAGA Performance

The overall success rate of CTTCB-LAGA was 94.3%. A mere 5.6% (18 out of 322) of biopsies resulted in an unsatisfactory diagnosis. Notably, the majority of biopsies demonstrated high efficiency, requiring only a single coaxial needle pass to procure sufficient tissue; specifically, 90.1% (173 out of 192) achieved this with just one pass. Two passes were necessary in 7.3% (14 out of 192) cases, while a minimal 2.6% (5 out of 192) required three or more passes, underscoring the impressive first-pass yield facilitated by LAGA guidance.

The median planned needle trajectory angle, as determined on CT images, was 18 degrees, ranging from 2 to 50 degrees. Remarkably, the median angle deviation between the planned and actual needle paths was a mere 2 degrees, highlighting the excellent adherence to the predetermined trajectory with LAGA assistance. The median depth from the pleura was 17.5 mm (range 4–50 mm) and 50 mm (range 10–120 mm) from the skin.

The complication rate for pneumothorax was 11.1%, with only one patient (0.3%) requiring chest tube insertion for a major pneumothorax (Table 2). The remaining cases had minor pneumothorax managed conservatively with oxygen or temporary aspiration. Pulmonary hemorrhage occurred after 9.0% of biopsies, but only three patients (1.0%) needed active treatment, such as embolization or transfusion for major bleeding. Hemoptysis was an uncommon complication, occurring in just 3.0% of cases. Only two patients (0.6%) had major hemoptysis requiring intervention like intubation. Minor complications like self-limited pain and cough were reported in 3.4% and 3.0% of patients, respectively. There was one reported case of air embolism.

Table 3 presents both univariate and multivariate analyses of factors influencing major outcomes in the context of 322 CTTCB-LAGA procedures. Univariate analysis revealed that a larger tumor size significantly increased the biopsy success rate, with a median of 25 mm for successful biopsies compared to 15 mm for failed biopsies (*p* = 0.009). A greater depth to the pleura was associated with a higher risk of pneumothorax, with a median of 60 mm for cases with pneumothorax compared to 50 mm (*p* = 0.018). Additionally, an increasing number of biopsy passes correlated with a higher incidence of hemorrhagic complications (*p* = 0.004).

In the multivariate regression analysis, tumor size emerged as an independent predictor of diagnostic success, with an odds ratio (OR) of 1.072 per mm (*p* = 0.022). Pleural depth stood out as the sole factor independently associated with pneumothorax, presenting an odds ratio of 1.019 per mm (*p* < 0.001). Other factors, such as patient age, gender, lung location, and biopsy needle trajectory angle, did not significantly impact major outcomes. Based on multivariate analysis, tumor size and pleural depth emerged as the principal factors influencing major biopsy outcomes. The LAGA system enabled high success rates regardless of experience.

## 4. Discussion

This study of 322 CT-guided lung biopsies assisted by the LAGA trajectory guidance system demonstrated excellent diagnostic success (94.3%) and safety outcomes, including low complication rates of pneumothorax (11.1%), pulmonary hemorrhage (9.0%), and hemoptysis (3.0%). The majority (90.1%) of procedures required only a single biopsy needle pass to obtain sufficient tissue. Tumor size emerged as an independent predictor of diagnostic success, while pleural depth was the main factor influencing pneumothorax risk. These findings highlight the potential benefits of trajectory guidance technologies like LAGA for optimizing this widely utilized diagnostic procedure.

The success rate and complications are pivotal considerations in CTTCB. Attaining diagnostic adequacy from CTTCB specimens is crucial for effective patient management. Comparative studies, such as that by Anderson et al., have highlighted higher success rates for core biopsy (93%) compared to fine needle aspiration (78%) [13]. Additionally, Diep et al. demonstrated a 90% success rate for next-generation sequencing with 16 and 18 G biopsy needles, surpassing the 30% success rate for 20 G needles [14]. In the genetic era, obtaining tissue is not solely for diagnosis but also imperative for genetic studies. Consequently, the utilization of large-bore core biopsy becomes essential, as reflected in our approach. Our biopsy success rate of 94.3% aligns favorably with the reported literature range of 67.6–95.0% (Table 4). Similarly, our observed complication rates compared favorably to 17 other published studies on conventional CT-guided lung biopsy spanning 1999 to 2022 (Table 4) [7,9,13,15,16,17,18,19,20,21,22,23,24,25,26,27,28], given that our research employed the largest bore size of biopsy needle among the reported literature. While the most commonly used core biopsy needle size is 18 G, we uniquely utilized a 16G needle, a choice shared only with the study by Heyer et al. [23]. The procedural approach resulted in a success rate of 94.3%, placing it among the highest reported in the literature. Specifically, Heyer et al. (95%) and Yuan et al. (94.8%) reported slightly higher success rates than ours [15,23]. Success rates across studies ranged from 67.6% to 94.8%, primarily from retrospective case series.

Examining pneumothorax rates, previous studies reported figures ranging from 11.1% to 65.1%, with chest tube insertion needed in 0% to 38.9% of cases. Notably, the pneumothorax rate in our study using CTTCB-LAGA was 11.1%, placing it at the lower end of the reported literature range. Furthermore, only 0.3% of cases required chest tube drainage for pneumothorax demonstrating a low rate of severe complications.

Considering hemoptysis rates, reported figures across studies ranged from 0.2% to 9.4%. However, many studies did not document hemoptysis rates. In our study, the observed 3.0% hemoptysis rate was comparatively low, indicating enhanced safety outcomes associated with the LAGA system compared to conventional freehand CT-guided lung biopsy, even with the use of the largest bore core biopsy needles.

Tumor size emerged as an independent predictor of success, consistent with prior studies indicating improved yields with more extensive lesions. Specifically, lesions measuring 10–15 mm represented the limitation for CTTCB, with smaller tumors potentially leading to biopsy failure [7,16]. Interestingly, depth to the pleura did not impact success in CTTCB-LAGA. This contrasts with some studies reporting lower diagnostic yields for deeper-seated lesions when using conventional freehand techniques [13], suggesting a potential advantage of LAGA guidance. The integration of our trajectory tract reference system has demonstrated high success, comparable to larger lesions biopsied with standard techniques. Emphasizing the importance of multiple samples and experienced cytopathology support, our findings illustrated the optimization of diagnostic yields from lung biopsy specimens.

Complications are not only significant for patients’ well-being but also bear implications for the timely and accurate attainment of the correct diagnosis. For instance, in cases where pneumothorax occurs before sufficient tissue is harvested, the tumor may deviate from the planned location, necessitating a reevaluation of the entire biopsy route after treating the pneumothorax. Similarly, severe hemoptysis leading to vital sign changes may prompt a temporary halt in the procedure [29]. Additionally, if lung hemorrhage occurs before tissue acquisition, the blood may obscure the tumor in the CT image, posing challenges in confirming the success of the biopsy. These complications can profoundly impact the procedural trajectory and emphasize the critical interplay between patient outcomes and diagnostic processes.

Pneumothorax stands out as the most prevalent complication of CT-guided lung biopsy, with reported incidence varying widely from 2.8% to 42% across published studies [7,16]. Our observed pneumothorax rate of 11.1% aligns favorably with this spectrum; the majority of pneumothorax was small and typically resolved spontaneously, requiring only observation or simple site-on aspiration [30,31]. Importantly, our low chest tube insertion rate of 0.3% demonstrated the generally minor nature of pneumothorax complications in our study. Previous studies reported that only 0–14.2% of cases required chest tube insertion for a symptomatic or enlarging pneumothorax [20]. The broad range in the reported pneumothorax rates likely stems from factors such as patient risk factors, biopsy techniques, needle size, image guidance, and the number of passes. Contrary to expectations, the incidence of pneumothorax is not necessarily higher in core biopsy compared to FNA [17]. Pleural depth emerged as an independent predictor of pneumothorax in our study, consistent with the understanding that deeper biopsies traverse more lung parenchyma [17,23,32,33]. Additionally, the approach to lesions contacting the pleura is a subject of ongoing debate in the literature. While some practices advocate for an approach that avoids lung parenchymal penetration to minimize the risk of pneumothorax [18,32,34], others suggest considering the transgression of normal lung tissue to biopsy subpleural lesions, aiming to reduce the number of pleural punctures [35]. This divergence in strategies accentuates the importance of individualized assessment based on specific lesion characteristics, patient factors, and the expertise with preferences of the interventionist. Our study’s findings contribute to this discussion, and we recognize that no single approach can be deemed universally superior in all clinical scenarios.

Small lesions, which may necessitate repeated punctures if missed, are also associated with increased pneumothorax risk [17,23,33]. The trajectory tract reference system minimizes this size effect by enhancing targeting precision. Notably, our study indicates that emphysema, often assumed to be associated with pneumothorax, does not show a significant correlation [25]. Even through emphysematous areas, the trajectory tracts were successfully negotiated, emphasizing the importance of careful planning and execution assisted by LAGA to reduce pneumothorax risks. Techniques such as coagulant patching along the biopsy tract offer additional strategies to mitigate pneumothorax risks [36]. While pneumothorax remains an expected potential complication, usually minor and not requiring intervention, careful planning and the use of the LAGA system can contribute to lowering pneumothorax rates through improved needle alignment and reduced passes.

Moving on to parenchymal hemorrhage, hemothorax, and hemoptysis, these potential complications of CTTCB can be alarming, arising from hemorrhage along the needle path due to pulmonary or intercostal vessel injury. The reported rates of hemoptysis range from 0.2% to 9.4%, and parenchymal hemorrhage rates vary widely from 3.9% to 10% in published studies, with severe hemoptysis being very uncommon [16,19]. Our study aligns with this spectrum, presenting a 3.0% overall hemoptysis rate, with only 0.6% requiring interventions such as intubation for major bleeding.

Risk factors for post-biopsy hemoptysis include smaller lesions, greater depth, and emphysematous lung parenchyma [19,23]. Large core needles are associated with more hemorrhage than FNA, emphasizing the importance of biopsy needle choice [21]. Protective measures, such as avoiding large vessels, sealing the tract with gelfoam or blood patches, and steering clear of repeated punctures through the aerated lung, further contribute to minimizing bleeding risk [32,34,36].

Despite the recognized risk, hemoptysis rarely leads to major clinical consequences or requires specific treatment post-biopsy. Our study accentuates the overall safety of CTTCB concerning hemoptysis risk, particularly with the assistance of LAGA to navigate the needle along less risky paths. Immediate interventions, such as trocar needle insertion and the application of sealant materials or diluted epinephrine through the coaxial needle, effectively manage bleeding in most cases. While hemoptysis is a common yet usually minor complication, ongoing research on techniques to prevent bleeding would undoubtedly enhance the safety profile of this widely employed biopsy method.

Systemic air embolism, although rare, represents a potentially catastrophic complication of CTTCB, with an estimated incidence of approximately 0.03–0.07% based on extensive case series [9,37]. The immediate detection of air emboli on CT scans is crucial in such cases. Notably, our study with CTTCB-LAGA did not observe any instances of air embolism. The proposed risk factors for systemic air embolism include needle entry into pulmonary venous structures, biopsy of consolidative lung lesions, and the repetition of puncturing and repositioning the needle [37,38]. Patient coughing or gasping during the procedure may also contribute to increasing intra-thoracic pressure. The consequences of air bubbles traveling to the left heart and onward to the coronary and cerebral circulations can be severe, leading to outcomes such as myocardial infarction, stroke, or even death.

To mitigate the risk of air embolism, protective measures in our practice include keeping the trocar of the coaxial needle within the tract as much as possible to prevent external air from entering the lung. Additionally, we aim to avoid vessels on the trajectory route, aided by our trajectory tract reference device. While the published rates of air emboli events are low, emphasizing the general safety of lung biopsy in this regard, it remains one of the most serious potential complications. Interventional radiologists must maintain a high level of suspicion and adhere to proper precautions. Our study suggests that the LAGA system provides excellent control over needle alignment, helping to avoid venous structures and minimizing the need for repositioning. Further innovations, such as methods to detect and aspirate intravascular air, could offer additional protective benefits against this rare but potentially devastating risk.

Numerous endeavors have been dedicated to enhancing the success rates of CTTCB while concurrently minimizing complications. Several variables, such as patient gender, age, comorbidities, emphysematous conditions, and tumor size, remain beyond our control. Among these, tumor size and depth are frequently cited as influential factors directly impacting the precision of the biopsy. The critical factor, however, is the precise targeting of the lesion.

Mathematically, the distance of deviation from the intended target (r) is intricately related to depth (D) and angle deviation (θ), expressed as $r=2\sin\left(\frac{1}{2}\theta \right)\times depth$ (Figure 2). Smaller tumors permit a smaller allowable radius for targeting error, while greater depth magnifies the impact of angular deviation. Previous studies predominantly highlight depth (D) as a factor influencing biopsy outcomes. Yet, the angular precision may wield even more influence, as deviation from the planned trajectory follows a sine curve where small θ changes provoke exponentially larger r, especially at greater depths (D). Missing the target often necessitates additional passes, potentially elevating the risk of complications.

Our prior work identified puncture frequency as a crucial complication factor, primarily determined by angular deviation and depth rather than the direct predictors [32]. While the depth of the tumor–pleura interface depends on tumor location, efforts are consistently made to select the shortest possible biopsy route. However, the deviation of the actual puncture angle from the planned trajectory is a factor that can be controlled, forming the focal point of our study using the Laser Angle Guide Assembly (LAGA) device. Our earlier research demonstrated that the LAGA system effectively minimizes angular deviations, maintaining them within a median range of only 1–3 degrees [10,11]. In a phantom trial, we showcased the LAGA’s efficacy in reducing both trajectory angle deviation and the number of required needle puncture passes [11].

Various tools have been employed to control biopsy needle angles, including fluoroscopy and cone-beam CT paired with an external laser for aiming [39]. However, the use of lead coats, glasses, and gloves for fluoroscopic CTTCB presents human factor engineering challenges and exposes the operator to radiation. Ren et al. asserted that cone-beam CT was superior to fluoroscopy, highlighting the potential benefits of a laser trajectory tract guide for cone-beam CT. Augmented reality, incorporating a virtual trajectory tract line, holds promise and has been tested in human trials [40].

When a tumor is located near the diaphragm, it tends to move with respiratory fluctuations, posing a challenge for accurate targeting during procedures. Respiratory motion correction, as proposed by Zhang [41], becomes essential in such scenarios. While respiratory gating is an effective tool for this purpose, its availability is limited, with respiratory gating primarily incorporated into PET-CT machines. Unfortunately, the high cost of PET-CT machines, not covered by our national insurance, makes them less accessible. Although many CT manufacturers offer EKG gating, there is a noticeable inclination away from respiratory gating. In our procedure, we address this challenge by placing a dot on the planned puncture point and rescanning, providing an additional opportunity to assess whether the natural respiratory movements might cause the target to shift out of focus. While attempting to instruct patients to hold their breath at the end of inspiration or expiration after some practice, the results have been disappointing. To navigate this issue, we conduct two pre-puncture scans and plan the procedure based on the end-inspiratory or end-expiratory phase of the patient’s quiet breathing, considering the available route with accessible intercostal space. This approach aims to optimize the chances of precise targeting despite the challenges posed by respiratory motion near the diaphragm.

This study’s reliance on a single-center retrospective analysis and the absence of a prospective comparison between procedures with and without Laser Angle Guide Assembly (LAGA) in our institution due to ethical constraints limit the direct evidence supporting the superiority of trajectory tract guidance. The comparison between procedures using LAGA and those without it was conducted solely in a phantom trial [11], as conducting CTTCB without a trajectory tract reference in our institution is ethically prohibited. The indirect nature of the inference, relying on mathematical models, a phantom trial, and comparisons with existing literature, introduces uncertainties regarding the utility of trajectory tract guidance in CTTCB. This aspect represents a valuable avenue for future research. A comparative analysis of procedure durations with and without the use of the LAGA system could provide further insights into the operational efficiency of the LAGA system, potentially underscoring its utility not only in improving safety outcomes but also in enhancing procedural efficiency. The study acknowledges emerging technologies, such as robotic arms for trajectory tract guidance, but their limited availability hinders a comprehensive assessment [42]. Ongoing trials, including the comparison trial between LAGA and cone-beam CT, are pending, and a thorough cost-effectiveness analysis is yet to be conducted, highlighting areas for future research and refinement of the study’s findings.

## 5. Conclusions

A trajectory tract reference such as LAGA demonstrates promise for enhancing CTTCB success, safety, and efficacy during this pivotal diagnostic procedure. The LAGA system presents itself as a valuable tool in the evolving landscape of CTTCB, showcasing potential advancements in trajectory guidance for improved patient outcomes.

## Figures and Tables

**Figure 1 diagnostics-14-00796-f001:**
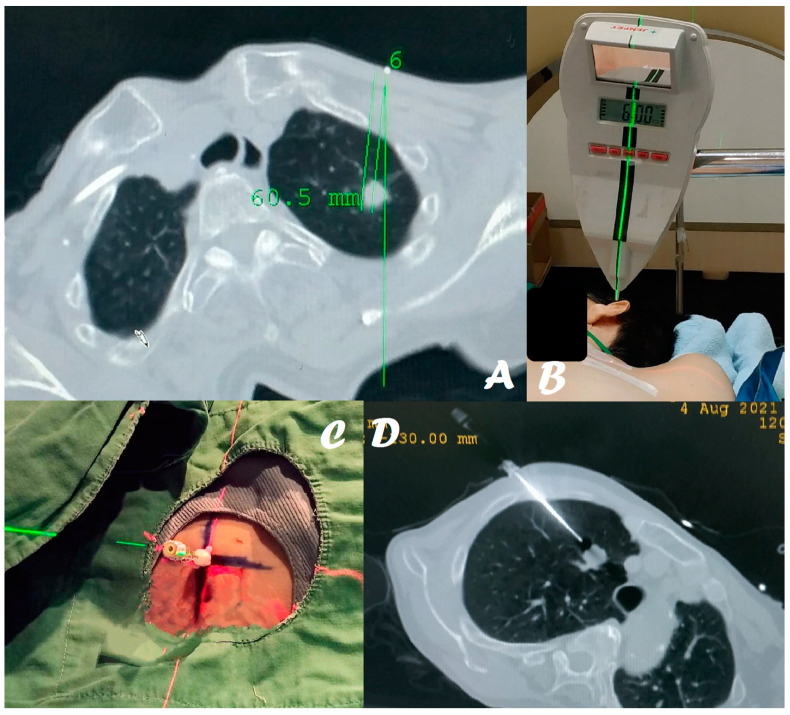
Procedure for CT-guided lung biopsy using the coaxial technique with Laser Angle Guide Assembly (LAGA). (**A**) Axial CT image used for procedure planning. A 15 mm lung tumor is noted in the left upper lobe (LUL). A 6° inward angle and 60.5 mm depth biopsy trajectory is planned. (**B**) The LAGA device is positioned according to the planned angle. Another green laser beam is projected from the caudal to the midline of LAGA along the planned biopsy trajectory to guide needle insertion. (**C**) After removing the LAGA device, the coaxial needle is inserted along the laser trajectory to the planned depth of 60.5 mm set by a stopper. (**D**) Post-biopsy axial CT image shows successful needle targeting of the LUL tumor.

**Figure 2 diagnostics-14-00796-f002:**
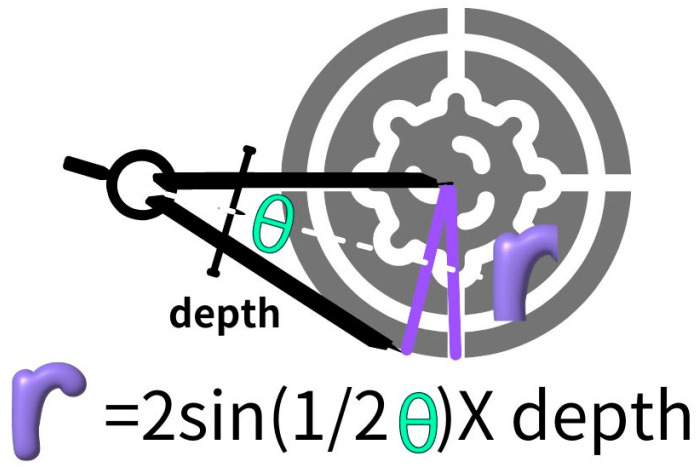
Relationship between biopsy trajectory accuracy and tumor size. This diagram shows the mathematical relationship between the planned biopsy trajectory angle (θ), depth deviation (Δd), and allowable targeting error (r) based on tumor radius. The formulas indicate that larger deviations in angle (θ) or depth (Δd) of a given tumor size will result in a biopsy needle missing the target tumor. This model can predict the maximum targeting errors that can be tolerated for successful sampling of lung lesions of a known diameter during CT-guided biopsy.

**Table 1 diagnostics-14-00796-t001:** Demographic and clinical characteristics of LAGA-assisted CT-guided lung biopsies (*n* = 322).

Characteristics	Category	*n*	%
Gender	Male	189	58.7
Age	Year, mean, SD	62.0	13.6
Procedure Success	Success	298	94.3
	Fail	18	5.7
Number of Punctures	1	173	90.1
	2	14	7.3
	≧3	5	2.6
Biopsy Location	RUL	92	28.5
	RML	19	5.9
	RLL	58	18.0
	LUL	60	18.6
	LLL	51	15.8
	Others	21	6.3
Patient Position	Supine	131	40.7
	Prone	162	50.3
	Decubitus	3	0.9
Number of Biopsy	1	11	3.4
Pieces	2	99	30.7
	3	128	39.8
	4	22	6.8
	5	8	2.5
	6	1	0.3
	7	1	0.3
Pathology	Primary	207	66.1
	Metastatic	16	5.1
	Benign	44	14
	Questionable	46	14.6
		Mean	SD
Pulmonary	FEV1%	79.7	18.2
Function	FVC%	83.9	18.0
	FEV1/FVC%	75.9	6.0
		Median	Range
Angle	Actual, °	18	2, 50
	Deviation, °	2	0, 4
Depth	Skin, mm	50	10, 120
	Pleural, mm	17.5	4, 50

Note: Some variables data missing *n* < 322.

**Table 2 diagnostics-14-00796-t002:** Complications and treatments from LAGA-assisted CT-guided lung biopsies (*n* = 322).

Complication	Treatment	*n*	%
Pneumothorax		36	11.1
	Drain	1	0.3
Lung hemorrhage		29	9.0
	Treatment	3	1.0
Hemoptysis		10	3.0
	Mild	8	2.4
	Treatment	2	0.6
Cough		11	3.4
Pain		10	3.0

**Table 3 diagnostics-14-00796-t003:** Factors associated with procedure success, pneumothorax, and hemorrhage in LAGA-assisted CT-guided lung biopsies.

Biopsy *n* = 322	Category	Unit	Success		Pneumothorax		Hemorrhage	
Gender	male/female	%	95.1/93.1	0.950	10.2/13.2	0.158	11.9/8.1	0.464
Age	year, yes/no	years, mean (SD)	66.0 (19.5)/64.0 (12.7)		65.9 (12.0)/64.7 (13.3)	0.131	69.2 (12.9)/64.2 (13.1)	0.775
Number of biopsies	1–5 11/96/126/2/8	%	82/95/95/100/7	0.470	18.2/8.3/15.1/9.5/0	0.143	54.4/7.2/9.4/4.8/12.5	0.004 *
Depth from skin	mm, yes/no	mm, median (IQR)	?50.0 (22.0)/40.0 (10.0)	0.098	60.0 (27.5)/50.0 (22.0)	0.018 *	50.0 (30.0)/50.0 (22.2)	0.238
	multivariate	OR (95%CI)	1.024 (0.996,1.052)	0.092	1.034 (1.006,1.063)	0.018 *		
Angle planned	yes/no	°, median (IQR)	18.0 (24.5)/14.5 (3.75)	0.281	23.0 (38.4)/18.0 (19.6)	0.577	7.3 (7.6)/18.0 (20.7)	0.319
Tumor size	yes/no	mm, median (IQR)	25.0 (15.0)/15.0 (2.5)	0.009 *	20.0 (22.0)/25.0 (15.0)	0.223	22.5 (26.2)/25.0 (14.5)	0.093
	multivariate	OR (95%CI)	1.072 (1.010–1.138)	0.022 *				
Lobe	RU/RM/RL/LU/LL	%	93/95/98/93/94	0.946	10.1/19.0/11.8/16.7/13.0	0.593	8.9/14.3/7.8/13.0/8.5	0.929
PT	yes/no	mean (SD)	10.6 (5.6)/10.2 (0.4)	0.840	10.2 (0.6)/10.6 (6.1)	0.599	10.3 (0.8)/10.6 (6.0)	0.723
aPTT	yes/no	mean (SD)	27.9 (3.5)/28.7 (1.5)	0.476	26.9 (5.9)/27.8 (3.5)	0.090	28.6 (4.2)/27.6 (3.8)	0.578
Platelet ×1000	yes/no	mean (SD)	260.0 (100.0)/234.7 (75.2)	0.430	241 (72)/258 (101)	0.247	233 (54)/259 (101)	0.059
Patient position	Supine/Prone	%	93.8/94.3	0.257	14.5/9.4	0.131	10.5/9.9	0.511

* signifies statistical significance (*p* < 0.05).

**Table 4 diagnostics-14-00796-t004:** Literature review of CT-guided lung biopsy procedure outcomes.

Author	Area	Year	N	Needle Size	Success %	Pneumothorax %	Intubation %	Hemorrhage%	Hemoptysis %
Tomiyama [8]	Japan	2006	9783			35.0			
Yuan [12]	China	2011	1014	core	94.8	12.9	1.5		4.0
Poulou [13]	Greece	2013	1000	core 35%	86.6	2.8		6.2	0.2
Loh [14]	Singapore	2013	399	FNA	91.9	34.8	3.0		3.2
Guimarães [15]	Brazil	2010	362	22G		11.1			1.9
Echevarria-Uraga [16]	Spain	2022	330	18G		24.2	5.7		9.4
Lin *	Taiwan	2023	322	16G	94.3	11.1	0.3	9.0	3.0
Saji [17]	Japan	2002	289	19G		26.2	14.2		
Anderson [18]	UK	2003	195	core/FNA	81.5				
Charig [19]	UK	2000	185	18G	93.5	25.9	2.2		7.0
Yaffe [20]	Israel	2015	181	FNA	93.6	26.5	2.2	3.9	6.0
Heyer [21]	Germany	2008	172	16G	95.0	26.0		10	
Kothary [22]	USA	2009	139	20G	67.6	34.5			5.0
Asai [23]	Japan	2013	102	18G	90.2	40.2	2.9		
Muehlstaedt [24]	Japan	2002	98	18G	94.0	21.0	2.0		
Lima [25]	Brazil	2011	97	25G	91.5	27.8	12.4		
Bungay [26]	UK	1999	88	22G		42.0	0.0		
Hirose [6]	Japan	2000	50	18G	94.0	42.0	12.0		

The sequence sorted according to the case numbers. * Lin 2023 is this CTTCB-LAGA study.

## Data Availability

The data supporting the findings of this study are available in the dataset titled “LAGA CT-guide lung biopsy” by Frank Lin, published in 2023 on Mendeley Data, version 1. The dataset can be accessed at https://doi.org/10.17632/w5cnn9zd37.1.

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
