# Peer review of "Optimizing Precision: A Trajectory Tract Reference Approach to Minimize Complications in CT-Guided Transthoracic Core Biopsy"

_diagnostics, 2024, doi:10.3390/diagnostics14080796_

Round 1

Reviewer 1 Report

Comments and Suggestions for Authors

As you use co-axial needle set, you need to better outline this circumstance in the paper; also type and diameter of biopsy needle should be mentioned;

2. The target mass co-axial access enables to use the valuable and effective  adjunct  - the puncture track sealing by gel-foam slurry; you either  do not use this possibility or this information is just missing - please, do the appropriate amendments

3. The major problem of lung mass correct aiming is the breathing motion - you need to outline better what is your way of this problem management; 

4. There is no comparison of procedure duration with and without LAGA system - would be interesting

Author Response

Author's Reply to the Review Report (Reviewer 1)

  1. As you use co-axial needle set, you need to better outline this circumstance in the paper; also type and diameter of biopsy needle should be mentioned;

Response 1: In response to your insightful suggestion, we agree that detailing the type and diameter of the biopsy needle will significantly enhance the manuscript by providing clarity and comprehensive insight into our methodology, potentially aiding the reproducibility of our study findings. To address Reviewer 1’s comment, we propose to add the following details in Section 2.2 (CTTCB-LAGA procedure), immediately after the mention of the coaxial needle set:

"Specifically, the biopsy procedures were performed using a 16-gauge x 15 cm TemnoTM coaxial needle set (BD, USA). The choice of a 16-gauge needle, larger than the commonly used 18-gauge needles for lung biopsies, was made to facilitate the acquisition of adequate tissue samples for both pathological diagnosis and genetic analysis, reflecting the evolving needs of genomic studies in the field of thoracic oncology."

  1. The target mass co-axial access enables to use the valuable and effective adjunct  - the puncture track sealing by gel-foam slurry; you either  do not use this possibility or this information is just missing - please, do the appropriate amendments;

Response 2: Thank you for your attention to the details concerning the use of adjunctive techniques, such as puncture track sealing with gel-foam slurry. We appreciate the opportunity to clarify this aspect of our methodology.

Regarding the use of gel-foam slurry for sealing the puncture track, we wish to inform the reviewer that this technique was not employed in our study. The primary reason for this omission is the unavailability of gel-foam slurry for this particular application within our practice in Taiwan at the time of conducting our research. Consequently, we did not utilize this adjunctive technique during the procedures described in our study.

In light of the reviewer’s valuable feedback, we propose to include a clarification within the manuscript to transparently communicate this to our readers. We suggest adding the following statement to the end of Section 2.2 (CTTCB-LAGA procedure), immediately after the detailed description of the needle insertion and biopsy process:

“In addition, it is important to mention that we did not use the adjunctive approach of sealing the puncture track with gel-foam slurry in our investigation, despite its known potential benefits in decreasing complications such as pneumothorax and bleeding. The decision was made due to the absence of gel-foam slurry for pulmonary treatments in Taiwan throughout the timeframe of our investigation.”

  1. The major problem of lung mass correct aiming is the breathing motion - you need to outline better what is your way of this problem management;

Response 3: Thank you for the reviewer’s guidance on accurately depicting our methodology regarding the management of breathing motion during lung mass biopsies. In our study, we did not rely on patient coaching for breath-holding due to the observed variability in lung positioning, which could lead to deviations in targeting accuracy. Instead, patients were asked to breathe normally. We utilized the end-inspiratory or end-expiratory phase for needle insertion, selecting the phase in which the target lesion was best visualized and most stable. This approach aimed to minimize the effects of breathing motion by synchronizing the timing of needle insertion with the natural respiratory cycle, thereby enhancing the precision of the biopsy procedure.

To incorporate this essential detail into our manuscript, we propose revising Section 2.2 (CTTCB-LAGA procedure) as follows:

"In addressing the challenge of respiratory motion during the biopsy, our strategy diverged from the conventional practice of instructing patients to hold their breath. Instead of relying on breath-holding, which could lead to variability and deviation owing to different capacities among patients, we asked them to breathe normally. We meticulously planned and executed needle insertion to coincide with the end-inspiratory or end-expiratory phase of the patient's normal breathing cycle, depending on which phase offered the greatest stability and visibility of the target lesion. This approach was facilitated by real-time CT imaging to determine the optimal moment for needle advancement, thus ensuring precise alignment with the patient's respiratory motion."

  1. There is no comparison of procedure duration with and without LAGA system - would be interesting.

Response 4: Given the reviewer’s feedback and our existing acknowledgment of the non-comparison of procedure durations in the "Discussion" section, we propose to enhance this section by explicitly stating the value of such comparison as an avenue for future research. This will provide readers with a clear understanding of potential areas where the LAGA system's impact could be further explored, beyond its efficacy and safety benefits.

To this end, we suggest adding the following sentences to the relevant part of the "Discussion" section:

"This aspect represents a valuable avenue for future research. A comparative analysis of procedure durations with and without the use of the LAGA system could provide further insights into the operational efficiency of the LAGA system, potentially underscoring its utility not only in improving safety outcomes but also in enhancing procedural efficiency."

By incorporating this statement, we aim to highlight the importance of assessing procedural efficiency as a critical factor in evaluating new medical devices and techniques, such as the LAGA system. It acknowledges the potential benefits of the LAGA system beyond safety and diagnostic yield, suggesting that future studies could yield valuable information regarding its impact on the workflow and time efficiency of CT-guided lung biopsies.

Thank you for guiding us to improve the clarity and depth of our manuscript. We are grateful for your constructive feedback and hope that our revisions meet your approval.

Reviewer 2 Report

Comments and Suggestions for Authors

A well written paper with a quite large population, which furtherly confirms some already well-established evidences regarding CT-guided transthoracic biopsy. Nevertheless, some points need improvements:

- Introduction, lines 44-46: the following statement "Bronchoscopic procedures, constrained to central lung regions, are less applicable, prompting a growing reliance on CT-guided approaches, especially for peripheral lesions. CT guidance facilitates precise needle placement, ensuring avoidance of damage to surrounding structures" is actually incorrect and needs to be reformulated: it is known and established that flexible broncoscopy biopsy demonstrates a high diagnostic yield for central lesions, being the preferred non surgical procedure for central lesions. To this purpose, citation of the following article "Baratella E, Cernic S, Minelli P, Furlan G, Crimì F, Rocco S, Ruaro B, Cova MA. Accuracy of CT-Guided Core-Needle Biopsy in Diagnosis of Thoracic Lesions Suspicious for Primitive Malignancy of the Lung: A Five-Year Retrospective Analysis. Tomography. 2022" is warmly advised.

- Some technical description about Laser Angle Guide Assembly (LAGA) system must be added in the introduction, in order to provide to the Reader some robust information concerning this device, which may be uncommon to several Institutions.

- Please integrate subpara. "2.2. CTTCB-LAGA procedure" with a thorough description of the CT acquisition protocol technical parameters, as well as technical specifications about the CT guidance system (e.g. manifacturer, MDCT, single source/double source, number of detector rows). If two or more CT system have been utilized, please provide the exact number of procedures performed under each CT.

- Discussion, lines 236-237: "Additionally, lesions contacting the pleura may necessitate an approach with no lung parenchymal penetration to minimize the risk of pneumothorax". The previous statement is actually still debated in Literature, since some Authors and up-to-date textbooks suggest considering transgressing normal lung to biopsy subpleural lesions in order to minimize pleural punctures (ref. Wible - DIAGNOSTIC IMAGING: INTERVENTIONAL RADIOLOGY, 3rd edition, Elsevier, page 597), thus needs to be downgraded, since it cannot be assumed as an absolute evidence.

  1.  

Author Response

Author's Reply to the Review Report (Reviewer 2)

A well written paper with a quite large population, which furtherly confirms some already well-established evidences regarding CT-guided transthoracic biopsy. Nevertheless, some points need improvements:

Comment 1: - Introduction, lines 44-46: the following statement "Bronchoscopic procedures, constrained to central lung regions, are less applicable, prompting a growing reliance on CT-guided approaches, especially for peripheral lesions. CT guidance facilitates precise needle placement, ensuring avoidance of damage to surrounding structures" is actually incorrect and needs to be reformulated: it is known and established that flexible bronchoscopy biopsy demonstrates a high diagnostic yield for central lesions, being the preferred non surgical procedure for central lesions. To this purpose, citation of the following article "Baratella E, Cernic S, Minelli P, Furlan G, Crimì F, Rocco S, Ruaro B, Cova MA. Accuracy of CT-Guided Core-Needle Biopsy in Diagnosis of Thoracic Lesions Suspicious for Primitive Malignancy of the Lung: A Five-Year Retrospective Analysis. Tomography. 2022" is warmly advised.

Response 1: Upon review, we agree that the original wording may inadvertently convey an incorrect perspective on the efficacy of bronchoscopic biopsy in diagnosing central lung lesions. Flexible bronchoscopy indeed remains a preferred non-surgical method for the evaluation of central lesions due to its high diagnostic yield in such cases.

To rectify this oversight and provide a more accurate representation of current diagnostic approaches, we propose the following revision to lines 44-46 of the Introduction:

"Bronchoscopic procedures are well-established for their high diagnostic yield in evaluating central lung lesions and represent the preferred non-surgical procedure for these cases. However, for peripheral lesions, where bronchoscopy might be less effective due to accessibility issues, CT-guided approaches have gained prominence. CT guidance facilitates precise needle placement, ensuring avoidance of damage to surrounding structures and extending the diagnostic capabilities to areas beyond the reach of bronchoscopy."

Additionally, we will incorporate the suggested citation to further support our statement and provide readers with a comprehensive overview of the diagnostic landscape for lung lesions

Comment 2: - Some technical description about Laser Angle Guide Assembly (LAGA) system must be added in the introduction, in order to provide to the Reader some robust information concerning this device, which may be uncommon to several Institutions.

Response 2: To address the reviewer’s feedback, we propose to insert a concise yet informative description of the LAGA system within the introduction. This addition aims to familiarize the reader with the device's purpose, functionality, and its significance in enhancing the precision of CT-guided lung biopsies. The following paragraph is suggested for inclusion after the overview of CT-guided biopsy techniques:

“The LAGA system comprises a portable laser unit that projects a beam to delineate the optimal needle trajectory based on pre-procedural CT images. This guidance allows for real-time adjustment of the needle's path, minimizing the risk of damaging surrounding structures and improving the likelihood of successful tissue sampling on the first attempt. The incorporation of the LAGA system into CT-guided biopsy procedures reflects a step forward in the pursuit of minimally invasive diagnostic techniques that prioritize patient safety and procedural efficiency.”

Comment 3: - Please integrate subpara. "2.2. CTTCB-LAGA procedure" with a thorough description of the CT acquisition protocol technical parameters, as well as technical specifications about the CT guidance system (e.g. manifacturer, MDCT, single source/double source, number of detector rows). If two or more CT system have been utilized, please provide the exact number of procedures performed under each CT.

Response 3: Based on the reviewer’s suggestion to provide a more detailed description of the CT acquisition protocol and the technical specifications of the CT guidance system used in our study, we have revised our approach to include specific details about the equipment and methodology employed:

“In this study, all CTTCB-LAGA procedures were conducted using the Philips CT Brilliance 64 system (Hi Tech International Group, Inc., Florida), equipped with a specialized lung biopsy protocol designed to optimize imaging quality while minimizing patient radiation exposure. The Philips CT Brilliance 64, a state-of-the-art multi-detector CT (MDCT) scanner, features 64 detector rows capable of providing high-resolution images essential for the precise localization and targeting of lung lesions. This system supports real-time CT imaging, facilitating accurate needle placement and adjustments during the biopsy procedure.

The lung biopsy protocol implemented on the Philips CT Brilliance 64 was carefully developed to ensure optimal visualization of the target lesion and the surrounding anatomical landmarks. Key parameters of the protocol included a slice thickness of 1 mm for detailed cross-sectional images, with tube voltage and current adjusted based on the patient's size and specific diagnostic requirements. This meticulous approach to CT acquisition allowed for the precise alignment of the biopsy needle with the target lesion, leveraging the advanced imaging capabilities of the Philips CT Brilliance 64 to enhance the safety and efficacy of the LAGA system during procedures.”

Comment 4: - Discussion, lines 236-237: "Additionally, lesions contacting the pleura may necessitate an approach with no lung parenchymal penetration to minimize the risk of pneumothorax". The previous statement is actually still debated in Literature, since some Authors and up-to-date textbooks suggest considering transgressing normal lung to biopsy subpleural lesions in order to minimize pleural punctures (ref. Wible - DIAGNOSTIC IMAGING: INTERVENTIONAL RADIOLOGY, 3rd edition, Elsevier, page 597), thus needs to be downgraded, since it cannot be assumed as an absolute evidence.

Response 4: In light of the reviewer’s feedback, we acknowledge the necessity to accurately reflect the complexity and nuances of current practices and the absence of consensus on the best approach to biopsy subpleural lesions. Therefore, we propose to amend the discussion in lines 236-237 of our manuscript to more accurately portray the variety of approaches and the ongoing debate regarding the optimal method for minimizing the risk of pneumothorax in such cases. Having added an appropriate reference, the revised text is proposed as follows:

“Additionally, the approach to lesions contacting the pleura is a subject of ongoing debate in the literature. While some practices advocate for an approach that avoids lung parenchymal penetration to minimize the risk of pneumothorax [18,32,34], others suggest considering the transgression of normal lung tissue to biopsy subpleural lesions, aiming to reduce the number of pleural punctures [35]. This divergence in strategies accentuates the importance of individualized assessment based on specific lesion characteristics, patient factors, and the expertise with preferences of the interventionist. Our study's findings contribute to this discussion, and we recognize that no single approach can be deemed universally superior in all clinical scenarios.”

We are grateful for the reviewer’s guidance, which has significantly enhanced the accuracy and depth of our discussion. Your feedback has been instrumental in improving the quality of our manuscript.
